# Bilosomes and Biloparticles for the Delivery of Lipophilic Drugs: A Preliminary Study

**DOI:** 10.3390/antiox12122025

**Published:** 2023-11-21

**Authors:** Maddalena Sguizzato, Francesca Ferrara, Nada Baraldo, Agnese Bondi, Annunziata Guarino, Markus Drechsler, Giuseppe Valacchi, Rita Cortesi

**Affiliations:** 1Department of Chemical, Pharmaceutical and Agricultural Sciences (DoCPAS), University of Ferrara, I-44121 Ferrara, Italy; sgzmdl@unife.it (M.S.); frrfnc3@unife.it (F.F.); nada.baraldo@edu.unife.it (N.B.); agnese.bondi@unife.it (A.B.); 2Department of Neurosciences and Rehabilitation, University of Ferrara, I-44121 Ferrara, Italy; annunziata.guarino@unife.it; 3Bavarian Polymer Institute (BPI), Keylab “Electron and Optical Microscopy”, University of Bayreuth, D-95440 Bayreuth, Germany; markus.drechsler@uni-bayreuth.de; 4Department of Environmental Sciences and Prevention, University of Ferrara, I-44121 Ferrara, Italy; giuseppe.valacchi@unife.it; 5Animal Science Department NC Research Campus, Plants for Human Health Institute, NC State University, Kannapolis, NC 28081, USA; 6Department of Food and Nutrition, Kyung Hee University, Seoul S02447, Republic of Korea; 7Biotechnology Interuniversity Consortium (C.I.B.), Ferrara Section, University of Ferrara, I-44121 Ferrara, Italy

**Keywords:** bile acids, nanovesicles, nanoparticles, SLN, NLC, drug solubility, nanotechnology, poorly water-soluble drugs, immunofluorescence, inflammasome

## Abstract

In this study, bile acid-based vesicles and nanoparticles (i.e., bilosomes and biloparticles) are studied to improve the water solubility of lipophilic drugs. Ursodeoxycholic acid, sodium cholate, sodium taurocholate and budesonide were used as bile acids and model drugs, respectively. Bilosomes and biloparticles were prepared following standard protocols with minor changes, after a preformulation study. The obtained systems showed good encapsulation efficiency and dimensional stability. Particularly, for biloparticles, the increase in encapsulation efficiency followed the order ursodeoxycholic acid < sodium cholate < sodium taurocholate. The in vitro release of budesonide from both bilosytems was performed by means of dialysis using either a nylon membrane or a portion of Wistar rat small intestine and two receiving solutions (i.e., simulated gastric and intestinal fluids). Both in gastric and intestinal fluid, budesonide was released from bilosystems more slowly than the reference solution, while biloparticles showed a significant improvement in the passage of budesonide into aqueous solution. Immunofluorescence experiments indicated that ursodeoxycholic acid bilosomes containing budesonide are effective in reducing the inflammatory response induced by glucose oxidase stimuli and counteract ox-inflammatory damage within intestinal cells.

## 1. Introduction

In recent years, the number of new molecules for which poor solubility in water represents the main obstacle to absorption has increased [1].

Indeed, the poor water solubility of a drug affects its bioavailability and the route of administration required to produce a therapeutic effect. Particularly, the water solubility of a drug is an important parameter to take into consideration when it comes to oral administration, being the preferred route for administering medicines. Notably, in order to be absorbed after oral administration, an active ingredient must pass into the solution in the aqueous gastrointestinal environment.

In this context, lipids represent a promising solution to this problem. It is known, in fact, that the intake of food rich in fat, together with lipophilic drugs, with poor solubility in water, can facilitate their absorption and therefore improve their bioavailability [1,2], depending on the numerous mechanisms related to the intake of food with a fair amount of lipid, including increased residence time in the intestinal lumen, increased biliary and pancreatic secretions, the stimulation of lymphatic transport, increased permeability of the intestinal wall, the inhibition of metabolic activities and efflux transporters and impaired mesenteric and hepatic blood flow.

With these concepts taken together, lipid-based delivery systems are of great interest to the oral administration of active ingredients, especially for those poorly soluble in aqueous environments. The goal of researchers is to exploit and optimize the ability of lipids to promote the absorption of an active ingredient in the intestine, possibly reducing the dose of drug that must be administered and, at the same time, limiting the dependence between the bioavailability of the drug and food intake [3].

Among the many types of lipid-based formulations that can be advantageously used to transport both hydrophilic and lipophilic drugs, in this preliminary study, lipid-based nanosystems, such as vesicles and nanoparticles containing different bile acids in their composition [4], namely, ursodeoxycholic acid (U), sodium cholate (C) and sodium taurocholate (T), are investigated and compared, with the aim of evaluating technological strategies to promote the bioavailability of poorly water-soluble drugs after oral administration. Indeed, bile salts are physiologically involved in the digestion of lipids; therefore, they can affect the absorption of poorly soluble drugs in different ways, especially if administered orally. Furthermore, many studies have reported that bile salt–phospholipid vesicles are able to transport the drug to the intestinal wall, leading to increased bioavailability [5,6,7,8] due to their additional ability to protect drug contents from the harsh environment of the gut [6,9,10].

As a model drug, budesonide was considered [11]. The choice of budesonide was made on the basis that it is considered the anti-inflammatory drug of “first choice” in the treatment of patients with mild to moderate inflammatory bowel disease (IBD) [12,13,14], but when administered orally, its absorption in the gastrointestinal tract is difficult, resulting in poor bioavailability [15,16,17], poor release at sites of intestinal inflammation and, therefore, low effectiveness [18]. Furthermore, it is necessary to take into account the high rate of metabolization and hepatic clearance [19]. Therefore, strategies able to improve patient compliance and gastrointestinal efficiency [20,21], such as bilosystems, become interesting in order to expand budesonide’s clinical use [21].

The aim of this study is to investigate the ability of formulations to improv the solubility and the intestinal absorption of budesonide, thanks to the presence of bile acids. To address this point, lipids for the production of bilosystems were selected on the basis of the most common components used in the preparation of traditional lipid-based nanosystems such as liposomes, SLN and NLC, as reported in the literature [22,23,24,25,26].

## 2. Materials and Methods

### 2.1. Materials

Phosphatidylcholine (Phospholipon 90 G, PC) was purchased from Lipoid GmbH (Ludwigshafen am Rhein, Germany), and cholesterol (CH), tristearin (stearic triglyceride), pluronic F-68, ursodeoxycholic acid (U), sodium cholate hydrate (C), sodium taurocholate (T), porcine lipase and budesonide were purchased from Merck KGaA (Darmstadt, Germany). Caprylic/capric triglycerides (Miglyol 812) were obtained from Sasol Germany GmbH (Witten, Germany). All other materials and solvents of high-performance liquid chromatography (HPLC) grade and of analytical grade were supplied by Carlo Erba (Rodano, Milan, Italy).

### 2.2. Preparation of Bilosomes

Bilosomes, with 25 mg/mL of final lipid concentration, were prepared via the hot hydration method. This technique involves the formation of a lipid film, consisting of PC, CH and bile acid (U, T and C) in a ratio of 4:1:1 mol/mol/mol, which was subsequently hydrated with an aqueous phase at 60 °C.

The various organic solutions were placed inside a flask (which was eventually padded with the drug) and subjected to forced evaporation, with the use of a rotary evaporator (70 bar, 200 rpm), until complete elimination of the organic solvent. The obtained thin lipid film was then hydrated with hot water, swirled (VELP scientifica srl, Usmate Velate, Italy) and subjected to 30 min of bath sonication at 40 °C [27,28].

### 2.3. Preparation of Biloparticles

Biloparticles were produced via hot homogenization followed by a sonication technique [29]. The lipid phase (5% of the total weight of the formulation) consisted of Tristearin, to which Tricaprin was added in the case of NLC. An aqueous solution of Poloxamer 188 (2.5% *w/w*) represented the remaining 95% of the preparation. Bile acids and the drug that would eventually be present were added in the lipid phase. Compositions of the different formulations are summarized in Appendix A.

The lipid phase was melted at 80 °C in a water bath. When the fusion was completed, the aqueous phase was rapidly added. Afterwards, the system was subjected to 1 min homogenization at 1500 rpm (IKA T25 digital ultra-turrax, VWR International srl, Milan, Italy). Subsequently the obtained emulsion underwent 15 min of ultrasound treatment (MICROSON Ultrasonic Cell Disruptor-XL MISONIX, VWR International srl, Milan, Italy) and the system was cooled down to room temperature.

### 2.4. Dimensional Analysis

Dimensional characterization of bilosomes and biloparticles was performed via Photon Correlation Spectroscopy (PCS) using a Zetasizer Nano S90 (Malvern Instr., Malvern, UK) equipped with a 5 Mw He/Ne laser with a wavelength output of λ = 633 nm. Samples were diluted 1:10 (*v/v*) with double distilled water up to 1 mL and poured in polystyrene cuvettes with four transparent faces. The measurements were thrice repeated at 25 °C at a 90° laser angle [30].

At the end of the analysis, a graph of the dimensional distribution of nanoparticles was obtained, giving the Z-Average and PdI. The first parameter represents the mean diameter of the nanoparticle population, while the second is the polydispersity index describing the dimensional homogeneity of the system (with <0.3 being the optimal value for solid nanoparticles) [31].

### 2.5. Encapsulation Efficiency Evaluation

To evaluate the encapsulation efficiency (EE) of the drug, the bilosystem was subjected to disruption to allow the passage of budesonide into the solution; then, its concentration was measured using a high-performance liquid chromatography (HPLC) system (Agilent Technologies, 1200 Series, Santa Clara, CA, USA) equipped with a UV detector and a 50 µL loop Rheodyne injection valve. The HPLC conditions for analysis are summarized in Table 1.

Particularly, 100 µL of bilosystem dispersion was mixed with 900 μL of HPLC mobile phase, and the system was kept under magnetic stirring for 3 h in an ice bath; then, the mix was filtered through 0.22 µm pore nylon filters and diluted 1:5 by volume with the mobile phase. Subsequently, each sample was injected twice in the chromatographic HPLC column and the drug concentration was calculated by comparison with a reference standard solution. The EE was calculated using Equation (1):EE = B/TB × 100(1)
where B is the amount of budesonide determined via HPLC and TB is the total budesonide amount used to prepare the formulation.

### 2.6. Equilibrium Dialysis Experiments

Budesonide release was evaluated via dialysis using either a nylon membrane (cutoff 10,000–12,000 Da) or a rat small intestine fragment and different receiving phases (i.e., fasted-state simulated gastric fluid, FaSSGF; fasted-state simulated intestinal fluid, FaSSIF; phosphate-buffered saline, PBS), whose composition is reported in Table 2.

Wistar rat intestine was used for permeation experiments (OpBA protocol codes CBCC2.50 and 373/2021-PR CBCC2.48). After removal from the animal, the intestine was washed with PBS, cut into 10 cm long fragments, stored at −20 °C until use and submerged in PBS for 12 h before use.

The membrane was filled with 1 mL of formulation, and then, closed at the ends with plastic pincers and submerged in 50 mL of receiving phase inside flasks kept on a tilting agitator (IKA LABORTECHNIK KS250basic). Samples of 500 µL were withdrawn from the receiving phase at predetermined time intervals during the 8 h of the experiment and filtered (nylon filters of 0.22 µm porosity) before HPLC analyses. After each collection, an equal volume of fresh receiving phase was reintegrated into the flask.

### 2.7. Cell Culture and Treatments

Human colorectal adenocarcinoma CaCo2 cells were cultured in high-glucose Dulbecco’s Modified Eagle’s Medium (Corning, Corning, NY, USA) supplemented with 10% FBS (Sigma-Aldrich, St. Louis, MO, USA), 100 U/mL penicillin, 100 μg/mL streptomycin (Gibco, Waltham, MA, USA) and 1% non-essential amino acids (Microtech cat. ACL006), as previously described [32]. All cell cultures were performed at 37 °C in 5% CO_2_ and 95% air. For the experiment, CaCo2 cells were treated for 24 h with different formulations containing budesonide, such as the ethanol solution (Sol B) and bilosomes (Bilos), at the indicated doses. Before the treatment, all the formulations were subjected to simulated digestion by incubating them for 30 min with FaSSIF. For the glucose oxidase (GO) experiments, cells were exposed to 0.5 U/mL of GO (G2133, Sigma Aldrich, Saint Louis, MO, USA) for 1 h, and then, samples were harvested 0 and 3 h after the end of GO insult for the following immunofluorescence and ELISA analyses.

#### 2.7.1. Cytotoxicity Study (MTT Assay)

To determine the non-toxic treatment dose, an MTT assay was performed on CaCo2 cells as previously described [32]. Briefly, CaCo2 cells were cultured in 96-well plates at a density of 1 × 10^4^ in 200 μL of media and treated for 24 h with the different formulations containing concentrations of budesonide ranging from 1 to 20 μg/mL. Upon removal of treatment, 110 μL of cell medium containing the MTT solution at a concentration of 0.5 mg/mL was added to each well and incubated for 4 h at 37 °C, 5% CO_2_. The insoluble purple formazan crystals were then dissolved in 100 μL of DMSO at 37 °C for 15 min and the solution absorbance was measured using a spectrophotometer at 570 nm, using 690 nm as a reference wavelength, and then, converted into percentage of viability.

#### 2.7.2. Immunofluorescence Staining

We seeded 1 × 10^5^ cells/mL of CaCo2 cells in 12-well plates and they were cultured for 24 h. Then, cells were pre-treated with the budesonide formulations (Sol B, Bilos) at a dose of 2.5 μg/mL and the respective unloaded formulations. After 24 h, cells were exposed to 0.5 U/mL of GO for 1 h, and then, collected right after the end of GO exposure (T0) and 3 h later (T3). Cells were washed in PBS, and then, fixed in 4% paraformaldehyde (PFA) in PBS for 10 min at room temperature. Cells were permeabilized with 0.25% Triton X-100 in PBS for 10 min at RT, and then, blocked in PBS-BSA 2% for 45 min at RT. Slides were incubated with primary antibody NLRP3 (NBP2-12446, Novus Biological, Littleton, CO, USA) 1:200 in 0.25% BSA/PBS and ASC (SC-514414, Santa Cruz Biotechnology Inc., Dallas, TX, USA) 1:100 in 0.25% BSA/PBS overnight at 4 °C. The samples were then incubated with fluorochrome-conjugated secondary antibodies (A11004 Alexa Fluor 568, A11008 Alexa Fluor 488) for 1 h at RT, and nuclei were stained with 1 μg/mL of DAPI (Sigma-Aldrich, Merck, Darmstadt, Germany) for 5 min. PermaFluor Aqueous Mounting Medium (TA-006-FM ThermoFisher Scientific, Waltham, MA, USA) was used to mount the coverslips onto glass slides, and the fluorescence signal was examined using a Leica light microscope (Leica, Wetzlar, Germany) equipped with epifluorescence at 40× magnification. Images were quantified using ImageJ software (ImageJ 1.53a, Wayne Rasband National Institute of Health, Bethesda, MD, USA).

#### 2.7.3. IL-1β ELISA Assay

An IL-1β ELISA kit (Cat. DY201-05, Novus Biologicals, Centennial, CO, USA) was employed to measure the levels of IL-1β in media of CaCo2 cells treated with budesonide formulations and collected 3 h post-GO exposure. Absorbance was measured using a spectrophotometer equipped with a filter of 450 nm, using 570 nm as a reference wavelength. IL-1β levels were expressed as pg/mL in culture media according to the manufacturer’s instructions.

### 2.8. Mathematical and Statistical Analysis

Concerning mathematical analysis, a graphical method is used to experimentally obtain the order of release. Indeed, the experimental data were plotted according to the equations of the different orders. Afterwards the coefficient of linear regression was determined and the coefficient of determination (R^2^) was then obtained. R^2^ values closest to 1 allowed us to obtain the order of reaction. Therefore, the release kinetics were determined via linear regression analysis of the in vitro release curves according to the best mathematical models expressing the kinetic release profile, namely, zero-order (cumulative amount (%) of drug released over time), first order (logarithmic cumulative amount (%) of drug released over time) and second order (Higuchi) (cumulative amount (%) of drug released over the square root of time).

The Shapiro–Wilk test was used to find the normality distribution of data, and statistical analysis was conducted using GraphPad Prism 9 (Version 9.4.1 (458), GraphPad Software Inc., La Jolla, CA, USA). Analysis of variance (1-way or 2-way ANOVA), followed by Tukey’s post hoc test, was used for each of the variables tested. Data are expressed as mean ± SD of duplicate determinations obtained in three independent experiments. The probability (*p*) value was considered non-significant statistically when *p* > 0.05, statistically significant when *p* ≤ 0.05 and highly significant when *p* ≤ 0.01.

## 3. Results and Discussion

### 3.1. Preformulation Studies

#### 3.1.1. Bilosomes

A preformulation study, which aimed to select and to identify the best characteristics of the formulation, in terms of stiffness of the lipid bilayer and its stability, was carried out on bilosomes composed of PC and ursodeoxycholic acid (U) (Bilos_PU) and bilosomes composed of PC, CH and U (Bilos_PCU), as described in Appendix A. As expected, Bilos_PCU dispersion was found to be more stable after production; therefore, a technological study was carried out on Bilos_PCU applying the same preparation protocols (i.e., Film hydration, Fh) with a few differences, such as hydration with room temperature solution (FhRT), hydration with room temperature solution followed by sonication (FhRTS), hydration with warm solution (FhW) and hydration with warm solution followed by sonication (FhWS).

The obtained bilosome dispersions were analyzed in terms of size via PCS, considering as dimensional parameters the mean diameter of the vesicles (Z-Average), the polydispersion index (PdI) and the scattering intensity (Intensity). The formulations, periodically analyzed to define their macroscopic and dimensional characteristics, indicated that FhWS is the most effective production process, leading to vesicles becoming stable over time (Appendix A). Indeed, the process carried out using room-temperature hydrating solution led to vesicles increasing in mean diameter and the formation of macroscopically visible aggregates one month after production. On the other hand, hydration conducted with warm solution (60 °C) seemed to favor the formation of stable and well-organized vesicles.

It is well known that the formation of the phospholipid bilayer is not a spontaneous process but requires energy. The phase transition of phospholipids varies according to the type of membrane composition, passing from an ordered “gel” phase to a more fluid and disordered “liquid-crystalline” phase; to reach this phase, characterized by greater molecular freedom of movement, an increase in temperature is needed to form the phospholipid bilayer [33]. Sonication seems to ensure stability over time, avoiding the formation of aggregates. In fact, in FhW preparations, the average diameter increases over time, whereas sonicated preparations show a gradual decrease over time and a gradual decrease in polydispersion to 0.3.

Therefore, the technological preformulation study indicated that warm hydration and subsequent sonication allow the membrane to undergo a thermotropic phase transition, giving greater stability and reducing aggregation phenomena.

The preformulative study was conducted on bilos_PC obtained in the presence of other bile acids, such as T or C. The preformulative study allowed us to identify the most efficient preparation technique and the most stable vesicular composition of the bilosomes. We then proceeded with the preparation of bilosomes containing the active principle budesonide for the following studies.

#### 3.1.2. Biloparticles

In the case of biloparticles, a preformulation study was performed to select the composition of lipid nanoparticles, namely, SLN (BilopS) and NLC (BilopN). Lipid nanoparticles were prepared by adding increasing amounts of bile acids, namely, ursodeoxycholic acid (BilopS_U), sodium cholate (BilopS_C) and sodium taurocholate (BilopS_T), with the aim of finding out the formulation leading to the best dimensional characteristics. Notably, direct proportionality between mean size (Z-Ave) and bile acid content for BilopS_C and BilopS_T was observed, which was not visible for BilopS_U. The type of bile acid induces large variability in terms of the size and polydispersity of nanoparticles at concentrations of 0.2%, 0.4% and 1.2%, while a content 0.8% of bile acid allowed for the maintenance of homogeneity in size and dispersion in all conditions (Appendix A). Therefore, on the basis of the obtained results, a content of 0.8% of bile acid was selected as the quantity to produce biloparticles for this study.

### 3.2. Production and Characterization of Bilosomes and Biloparticles Containing Budesonide

#### 3.2.1. Bilosomes

Taking into account the results of the preformulation studies, bilosomes were produced in the presence of budesonide, and the vesicles were characterized in terms of morphology, size distribution and encapsulation yield. As an example, in Figure 1 images of empty and budesonide-loaded Bilos_PCU are reported.

The PCS data obtained for the produced bilosomes were compared with those of the corresponding empty preparations. Figure 2 shows the Z-Average and PdI values of the produced bilosomes that were unloaded (Bilos_PCU, Bilos_PCT, Bilos_PCC) and loaded with the drug (Bilos_PCUB, Bilos_PCTB, Bilos_PCCB). In particular, Bilos_PCUB’s mean size is 410.8 nm, comparable to that of Bilos_PCU (419.9 nm). Bilos_PCTB (338.4 nm) and Bilos_PCCB (150.4 nm) instead exhibit a much smaller mean diameter than the corresponding empty bilosome dispersions (i.e., Bilos_PCT 458.5 nm and Bilos_PCC 320.82 nm).

Observing the PdI data, it can be seen that budesonide-loaded U bilosomes show slightly different polydispersity values compared to the corresponding unloaded formulation, with 0.31 for Bilos_PCUB and 0.35 for Bilos_PCU. Bilos_PCT (0.48) and Bilos_PCC (0.43) instead present higher PdI values than the corresponding formulations containing budesonide (i.e., Bilos_PCTB 0.35 and Bilos_PCCB 0.32).

From the analysis of the reported results, it emerges that the dimensional variation can be attributed to the presence of the drug. This behavior is more evident for bilos_PCCB as compared to bilos_PCUB or bilos_PCTB. We tentatively hypothesized that the assembling of the drug within the bilayer could be responsible for vesicle size reduction due to the interactions among the lipophilic moieties of phospholipids and bile acids, on the basis of some studies demonstrating that increasing amounts of cholesterol (a steroid molecule similar to budesonide) in the liposome bilayer improves particle uniformity (PDI) [25,34,35], and that a synthetic budesonide derivative assembled into liposomal nanovesicles for oral delivery shows favorable colloidal stability in different media (i.e., PBS, FASSGF, FASSIF) [21].

Indeed, the presence of a budesonide steroid nucleus with hydrophilic groups in positions 11β and 17β confers an amphipathic nature to the molecules, allowing the intercalation of the drug within the membrane, possibly improving the characteristics of the bilayer and making the size more homogeneous and monodisperse.

Concerning the efficiency of drug encapsulation (encapsulation efficiency, EE%), an essential parameter to indicate how much drug has actually been encapsulated inside the nanocarrier, it was calculated as the quantity of drug present in the formulation with respect to the drug content weighed for the preparation. The analysis was performed via HPLC the day after production. The percentage yield of the drug was calculated using the mathematical equation described in the Materials and Methods section.

It was found that all the formulations present high encapsulation yields. Notably Bilos_PCUB displays the highest percentage yield (92.76 ± 7.12), while both Bilos_PCTB (88.8 ± 17.9) and Bilos_PCCB (88.7 ± 21.9) display superimposable values. This behavior corroborates studies describing that the film hydration method, leading to multilamellar vesicles, is able to highly incorporate lipophilic drugs due to their intercalation between the alkyl chains of phospholipids [36,37,38,39].

#### 3.2.2. Biloparticles

Biloparticles with compositions obtained from the preformulation study were loaded with budesonide, and then, characterized. The biloparticles’ morphology was investigated via cryo-TEM, and the exemplificative images of budesonide-loaded BilopS_U and BilopN_U are reported in Figure 3. No differences in terms of morphology are evident in drug-loaded bilopS and bilopN with the presence of different bile acids (i.e., S and T).

In general, the three-dimensional particles are projected in a two-dimensional way. Concerning bilopS, elongated circular platelet-like crystalline particles and electron-dense “needle-like” structures, when viewed edge-on, can be observed due to the tilted position of the particle. On the other hand, bilopN’s shape appears roundish, discoid from the top view, or more electron-dense and rod-like from the edge-on view. Some inner ultrastructures are visible, showing several layers [40].

Figure 4 shows the dimensional characteristics of the produced biloparticle dispersions loaded with budesonide. In terms of size, BilopS usually displays a larger mean diameter than BilopN. Additionally, BilopN shows a more homogenous size distribution compared to BilopS; in fact, the difference between the PdI of BilopS and BilopN is very small.

The encapsulation efficiency of budesonide of both BiolpS and BilopN is summarized in Table 3. In general, BilopS shows higher encapsulation efficiency and lower drug loss in the aggregate compared to BilopN. For both types of biloparticles, the increase in encapsulation efficiency follows the order Bilop_U < Bilop_C < Bilop_T.

### 3.3. In Vitro and Ex Vivo Studies

The release kinetics of budesonide from bilosomes and biloparticles were studied in vitro via equilibrium dialysis. In particular, two different types of membranes were used, a synthetic one (nylon cutoff 10,000–12,000 Da) and a natural one consisting of a portion of the small intestine of a Wistar rat. Furthermore, two different receiving solutions were considered: FaSSGF, able to simulate gastric fluid, and FaSSIF, able to simulate intestinal fluid (Table 2). The experimental model is described in the Materials and Methods section.

The in vitro study attempts to mimic in vivo oral administration. Once administered, the formulation encounters environments that differ profoundly in terms of pH, the presence of enzymes and electrolytes, fluid viscosity and mucous surface characteristics. All of these factors can affect drug absorption; therefore, through equilibrium dialysis, it is possible to investigate the behavior of the drug by mimicking gastrointestinal transit using artificial gastric and intestinal fluid, with their specific characteristics in terms of pH and composition.

The experiment was conducted for all the bilosystems containing budesonide, namely, Bilos_PCUB, Bilos_PCTB, Bilos_PCCB, BilopS_UB, BilopN_UB, BilopS_CB, BilopN_CB, BilopS_TB and BilopN_TB, which were then compared with a dispersion of the drug in FaSSGF or FaSSIF (Susp B) and with an ethanol solution of the drug (Sol B).

The release kinetics obtained for the different formulations are represented in Figure 5. The left column displays the results of experiments with FaSSGF as the receiving phase, whereas the column on the right reports the release in the FaSSIF receiving phase.

From the obtained results, it is possible to notice that, both in FaSSGF and FaSSIF, budesonide is released from the bilosystems more slowly than from Sol B, which reaches 100% of release after the third hour, with no significant differences between gastric and intestinal environments. On the other hand, the drug released from Susp B (either in FaSSGF or FaSSIF) reaches 4% after 24 h. In the same time interval, most of the bilosystem formulations release a maximum of 20% of the drug. Indeed, comparing the percentage of budesonide released by Bilos_UB in the gastric fluid, it reaches a maximum of 37%, while in the intestinal fluid, the maximum release is 20%. This indicates that in an acidic environment, the presence of U in the bilosome composition causes a greater release than in a basic environment and in the presence of lipase.

The pH-sensitive release may enhance the targeting ability of the drug-loaded bilosomes following the gastrointestinal absorption of intact vesicles. However, this requires further in vivo confirmatory investigations.

Conversely, Bilos_PCTB and Bilos_PCCB in the gastric environment (Figure 5a) show approximately the same trend, with a release of 30% after 8 h and 37% after 24 h. In the intestinal environment (Figure 5b), bilos_PCCB shows a release of 37% after 8 h, which remains constant even at the twenty-fourth hour. In the case of Bilos_PCC, a 25% release and a maximum of 28% after 24 h is evident, lower than Bilos_PCTB.

This experiment indicates that Bilos_PCTB is the formulation able to release higher quantities of budesonide in the intestinal and gastric environments (40% release). Bilos_PCCB, on the other hand, keeps the release superimposable in both receiving phases, increasing by 7% in the gastric fluid after 24 h. In the intestinal environment Bilos_PCUB is lower in terms of percentage release, while in the gastric environment, it surpasses the other preparations up to the eighth hour, and then, remains constant (37%).

The comparison of biloparticle release curves with those of budesonide dispersed in FaSSGF and FaSSIF highlights a significant improvement in budesonide’s passage into the aqueous solution when carried by biloparticles. Budesonide is a lipophilic molecule with poor water solubility and cannot solubilize on its own in FaSSIF and FaSSGF, leading to a very low percentage of release even after 24 h (only 3% of the active ingredient released), while its encapsulation in biloparticles increases the release up to 70%.

In addition, it is possible to observe some differences in the release of bilopS and bilopN, depending on the type of bile acid and on the receiving phase. Specifically, for both bilopS and bilopN, greater release percentages were obtained in FaSSIF than in FaSSGF. BilopN_UB, bilopN_CB and bilopN_TB are able to release budesonide faster than the corresponding bilopS in FaSSGF and FaSSIF.

Among bilopN, bilopN_CB shows the highest release kinetics, especially in FaSSGF. In particular, after 24 h, 72% of budesonide was released in FaSSIF and 67% in FaSSGF. Concerning bilopS, BilopS_UB and BilopS_CB show the highest release kinetics both in FaSSGF and FaSSIF, while BilopS_TB has the slowest release; in fact, after 24 h, only 53% of the drug was released in the intestinal environment and 45% in the gastric environment. Percentages greater than 60% were achieved for all other formulations after the same time interval.

Subsequently, in order to have information about how budesonide could cross the intestinal wall, an experiment using an ex vivo model was carried out. The experiment was performed using a rat small intestine fragment as a dialysis tube, and phosphate saline buffer (PBS) as the receiving phase. The bilosomes and biloparticles, namely, Bilos_PCUB, BilopS_UB, BilopN_UB, Bilos_PCCB, BilopS_CB, BilopN_CB, Bilos_PCTB, BilopS_TB and BilopN_TB, diluted 1:1 by volume with FaSSIF, were inserted into the intestinal tube, which was suitably closed at the ends, to mimic the in vivo conditions. As a control, a budesonide suspension in FaSSIF (Susp B) was used.

In general, in agreement with the literature [41], it was found that budesonide loaded into bilosystems reveals remarkable improvement in drug intestinal transport compared to simple budesonide aqueous suspension. Notably, from the analysis of Figure 6a, Bilos_PCUB and Bilos_PCTB display almost the same trend, increasing the release up to 20% at 8 h. The release from Bilos_PCCB is higher than the other Bilos, reaching a release of about 30% of the total drug content.

By comparing bilopS and bilopN (Figure 6b,c), it is evident that both can enhance budesonide intestinal absorption compared to the active ingredient in suspension, which does not exceed 5% of release after 8 h. Notably, for bilopS, after 3 h, at least 40% of budesonide is released from the biloparticles, reaching the 60% after 8 h. Moreover, the presence of different bile acids seems to not affect the release of the drug.

Regarding bilopN (Figure 6c), the expected increase in Budesonide release involves only bilopN_U and bilopN_T.

From the comparison between bilopS and bilopN (Figure 6b,c), the drug release from bilosystems containing U or T displays similar trends, while bilopN_C is characterized by slower release kinetics than the corresponding bilopS_C.

These results indicate that the bilosystem (either bilosome or biloparticle) is able to increase the solubility and to control the release of the incorporated drug. Furthermore, these data revealed the significant dominance of the bilosystem to enhance the intestinal passage of budesonide, possibly due to the ability of these nanocarriers to improve the intestinal permeability of the encapsulated drugs [42,43,44,45,46,47,48,49,50,51,52,53,54,55,56].

#### 3.3.1. Mathematical Analysis

It is well known that the passage of a molecule from a donor compartment to a recipient increases until it reaches equilibrium, with the transit rate depending on the physico-chemical properties of the drug, the nature of the membrane and the gradient concentration between the two compartments.

On the basis of Fick’s laws, it is possible to obtain the release rate of the drug described by kinetic models in which the drug is released as a function of time [57]. Particularly, both the empirical and mechanistic mathematical models based on the Fick’s laws are usually employed to describe the drug release from a drug delivery system [58]. The main types of kinetics are the zero-, first- and second-order (or Higuchi).

The zero-order equation (Table 4) describes a release directly proportional to time, such as that from matrices with low solubility and osmotic systems. Notably, the carrier enables the release of the same amount of drug per unit of time, with the release being independent of its concentration in the system. On the other hand, the first-order equation (Table 4) relates to a drug release rate depending on the concentration of the drug remaining within the system, typical, for instance, of the release from a porous matrix.

Concerning the second-order equation (Higuchi model), being substantially derived from Fick diffusion law [59], a direct relationship between the drug release rate from the matrix and the square root of time is described as typical of a homogeneous non-erodible granular matrix. Particularly, the Higuchi equation (Table 4) [60] refers to systems in which a) the drug concentration at the beginning is greater than the drug solubility, b) the matrix is a thin film that is not erodible or degradable, c) the size of the drug particle is lower than the film thickness and d) the diffusivity of the drug does not depend on space and time.

As shown in Table 4, the second-order (Higuchi) model expresses the highest value of the R^2^ coefficient for all types of bilosome preparations when the synthetic membrane (nylon) is used for the release. On the other hand, the bilosome kinetics obtained from a natural membrane (rat intestine) follow the zero order.

Concerning biloparticles, different behavior was found. Indeed, budesonides’ release kinetics followed the first order when a nylon membrane was used for the experiment, whereas in the case of intestinal membrane, a second-order release was detected.

This different behavior between bilosomes and biloparticles is ascribable either to the nature of the nanosystem or to the type of interaction with the membrane.

Indeed, when the vesicular systems are poured into a nylon bag, the bilosomes are stable, acting as a non-degradable system, and the release is independent of space and time. On the other hand, when the vesicles are in contact with the intestinal tube, they interact with the membrane and possibly fuse with it, allowing for a concentration independent of the release by the system.

Regarding biloparticles, the transition from the first to the second order can be due to the solid state of the particles compared to the more fluid vesicles. Notably, when poured in the synthetic membrane, it seems that the drug release from biloparticles mainly depends on the structural conformation of the matrix and the drug concentration loaded within the particles, whereas when the release is carried out within the intestinal membrane, the system becomes more stable and less erodible, possibly due to a firm interaction with the mucous on the tissue surface.

#### 3.3.2. Cell Studies

On the basis of the encapsulation efficiency of the drug (Table 3) and the drug content stability over time, Bilos_PCU and Bilos_PCUB were selected for the experiments on cultured cells.

##### MTT Assay

To select the non-toxic treatment dose to be used in the experiments, CaCo2 cells were treated for 24 h with formulations containing or not containing budesonide at a concentration ranging from 1 to 20 μg/mL and subjected to an MTT assay. As shown in Figure 7, cells treated with the solution containing budesonide (Sol B) at a dose above 2.5 μg/mL displayed a decrease in viability (less than 80%) compared to untreated controls cells, whereas Bilos_PCUB formulations did not significantly affect cell viability at any selected doses. Considering that the aim of this study was to compare whether Bilos_PCU is a more effective vehicle for budesonide with respect to the solution, we selected 2.5 μg/mL as the safe treatment dose to use in further analysis.

##### Immunofluorescence NLRP3-ASC and ELISA IL-1β

Budesonide is a corticosteroid used as an anti-inflammatory drug in many gastric and intestinal pathologies [11]. Inflammasomes are multiprotein complexes of the innate immune systems that are currently widely studied due to their role in mediating the inflammatory response in several tissues [61].

These inflammatory pathways can be activated by different stimuli, including pathogens and oxidative stress mediators such as reactive oxygen species (ROS). In particular, stimuli able to induce oxidative stress reactions have been shown to promote the activation of inflammasome pathways, and thus, the inflammatory response [62].

For instance, NLRP3, one of the most well-known inflammasomes, has been shown to mediate the inflammatory response in intestinal disorders via an oxidative stress mechanism [63,64,65]. Thus, to understand whether the NLRP3 inflammasome could be activated in human intestinal cells in response to an oxidative stimulus, and to evaluate whether formulations containing budesonide could prevent an oxidative stress-related inflammatory response, as already evidenced [66], we triggered CaCo2 cells with the oxidoreductase enzyme glucose oxidase (GO), able to induce the production of hydrogen peroxide (H_2_O_2_) via glucose oxidation as previously described [67].

Our results show that GO exposure could induce an inflammatory state in CaCo2 cells, represented by increased expression levels of NLRP3 and ASC right after the end of exposure (Figure 8).

Notably, pre-treatment with Bilos_PCUB could significantly prevent the induction of NLRP3 and ASC upon GO insult, whereas Sol B could only partially prevent the damage. Interestingly, CaCo2 cells pretreated with Sol B and not exposed to GO also presented an increase in inflammasome component expression, suggesting that when budesonide is not conveyed/included in a delivery system, it can be irritating for cells.

When activated, the inflammasome pathway results in the production and maturation of inflammatory cytokines such as Interleukin-1β (IL-1β) that can propagate the inflammatory response. Thus, we measured the levels of IL-1β in media of CaCo2 cells exposed to GO and pre-treated with budesonide formulations.

As shown in Figure 9, we found increased levels of IL-1β 3 h after the end of GO exposure. These data confirm the activation of the inflammasome and the establishment of an inflammatory response. When treated with both Sol B and Bilos_PCUB, the levels of IL-1β were completely restored to the basal state. Of note, Bilos_PCUB was even more effective in reducing the inflammatory response induced by GO stimuli compared to Sol B, confirming that this delivery system can serve as a better vehicle for budesonide to counteract ox-inflammatory damage within intestinal cells. As a matter of fact, the interplay between oxidative and inflammatory mediators can promote a phenomenon called ox-inflammation [68], which often occurs in many conditions, including intestinal disorders [69].

## 4. Conclusions

This study proposes and compares different types of bile acid-based nanosystems, namely, bilosomes and biloparticles, as delivery systems to improve the solubility and to preserve the activity of lipophilic drugs. For instance, at present, bilosomes are used to deliver numerous biological macromolecules [45,70,71,72]. It was found that bilosystems are able to efficiently encapsulate the drug while preserving its properties. However, the influence of different bile acids (i.e., U, C, T) on the performance of the produced bilosystem needs to be studied in depth to try to understand their contribution to specific drug release properties. This study, in fact, being preliminary, focused mainly on macroscopic evidence of the different behaviors of the produced bilosystems. Subsequent studies will be carried out that try to delve deeper into the reasons for the different behaviors expressed by the different bile acids used. In addition, our encouraging results, showing the efficient antioxidant effect of the complete restoration of IL-1β to the basal level after Bilos_PCUB treatment, suggest the possibility of further investigating the use of bilosystems to counteract ox-inflammatory damage within intestinal cells, especially for those intestinal pathologies correlated with the development of an oxidative stress status.

## Figures and Tables

**Figure 1 antioxidants-12-02025-f001:**
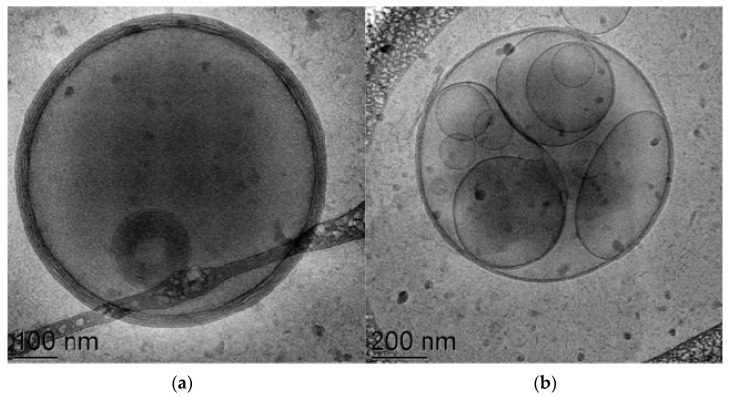
Cryo-TEM images of Bilos_PCU without (**a**) and with (**b**) budesonide prepared via a warm film hydration method followed by sonication (FhWS).

**Figure 2 antioxidants-12-02025-f002:**
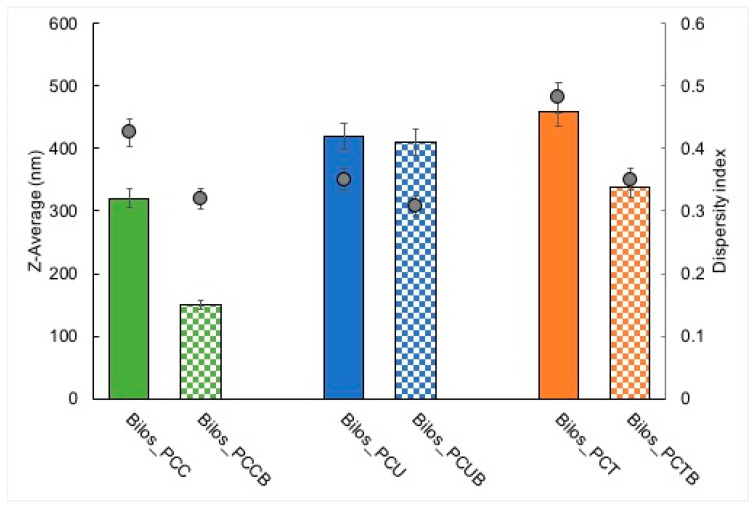
Dimensional parameters of bilosomes with (squares) and without (plain) budesonide, expressed as average diameter (histograms) and dispersity index (circles). Values are the mean of 3 batches ± s.d.

**Figure 3 antioxidants-12-02025-f003:**
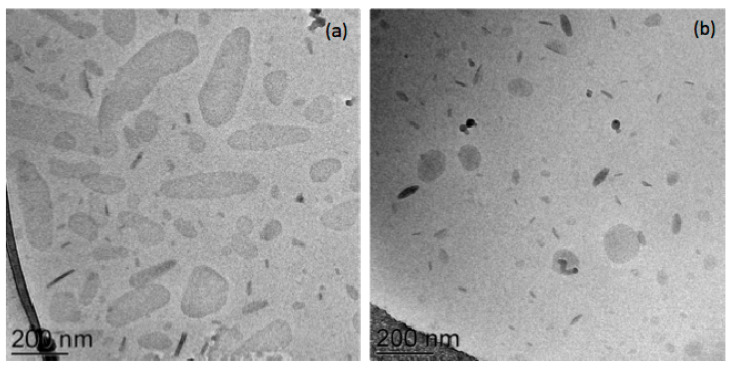
Cryo-TEM images of BilopS_UB (**a**) and BilopN_UB (**b**).

**Figure 4 antioxidants-12-02025-f004:**
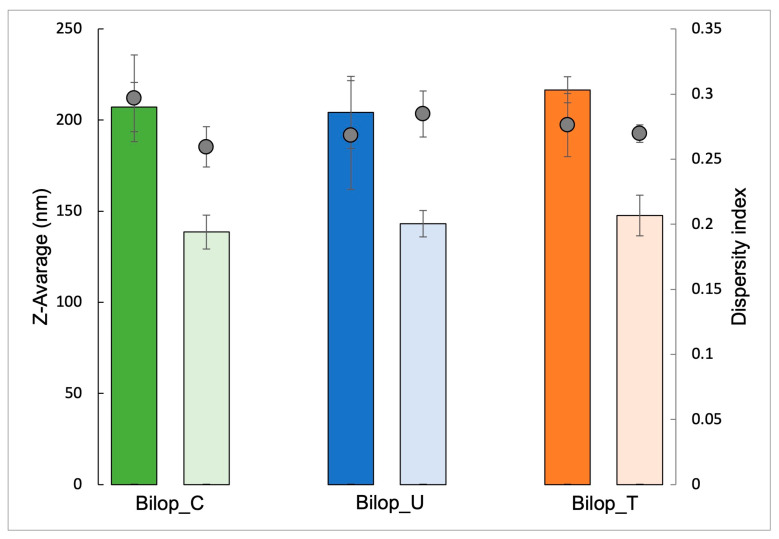
Mean diameter (Z-Average, histograms) and polydispersity (PdI, circles) of BilopS (dark color) and BilopN (light color) containing different bile acids, as determined via PCS. Each value is the average of 5 batches ± s.d.

**Figure 5 antioxidants-12-02025-f005:**
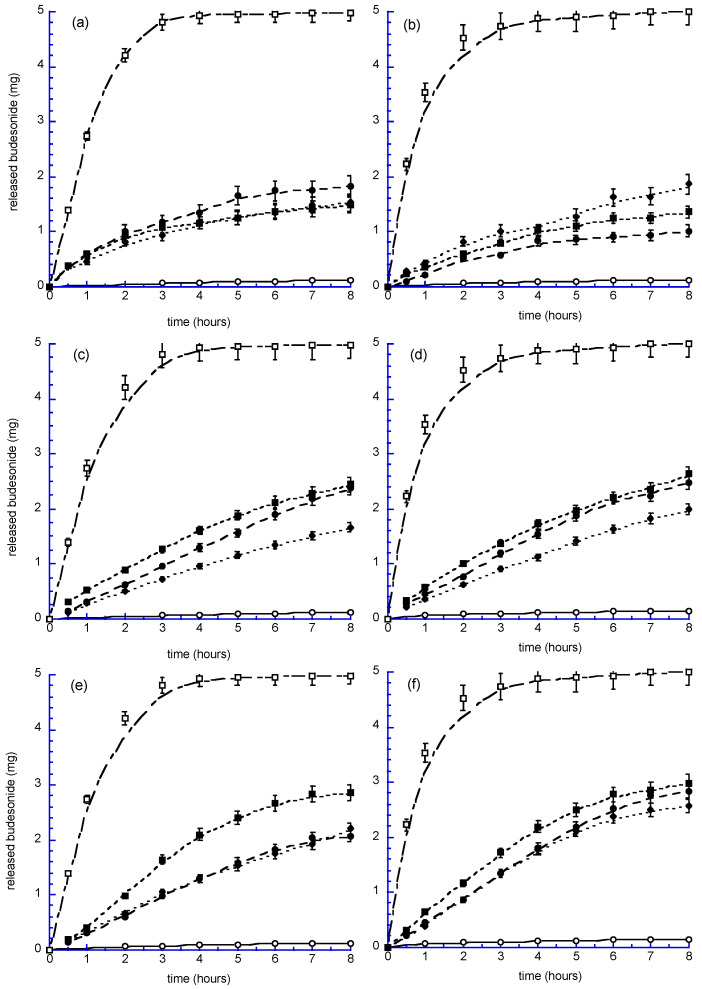
Budesonide release kinetics in FaSSGF (**a**,**c**,**e**) and FaSSIF (**b**,**d**,**f**) from Bilos (**a**,**b**), BilopS (**c**,**d**) and BilopN (**e**,**f**). Bilos_PCUB, BilopS_UB and BilopN_UB: closed circle; Bilos_PCCB, BilopS_CB and BilopN_CB: closed square; Bilos_PCTB, BilopS_TB and BilopN_TB: closed diamond; For comparison, the releases from Susp B in FaSSGF/FaSSIF (open circle) and Sol B (open square) are also reported. Each datum is the average of 2 experiments ± s.d.

**Figure 6 antioxidants-12-02025-f006:**
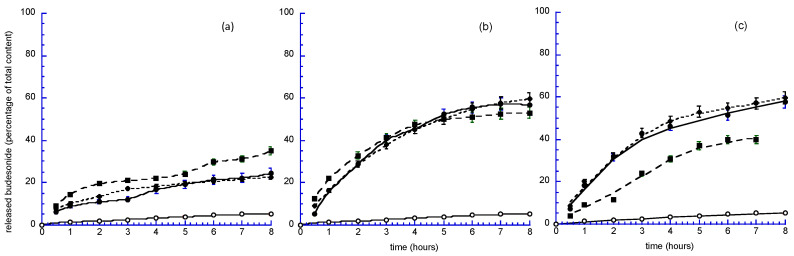
Budesonide releases kinetics through rat small intestine from Bilos (**a**), BilopS (**b**) and BilopN (**c**) suspended in PBS/FaSSIF (50:50 *v*:*v*). Bilos_PCUB, BilopS_UB and BilopN_UB: closed circle; Bilos_PCCB, BilopS_CB and BilopN_CB: closed square; Bilos_PCTB, BilopS_TB and BilopN_TB: closed diamond; The release from FaSSIF Susp B (open circle) is also reported for comparison. Each value is the mean of 3 experiments ± s.d.

**Figure 7 antioxidants-12-02025-f007:**
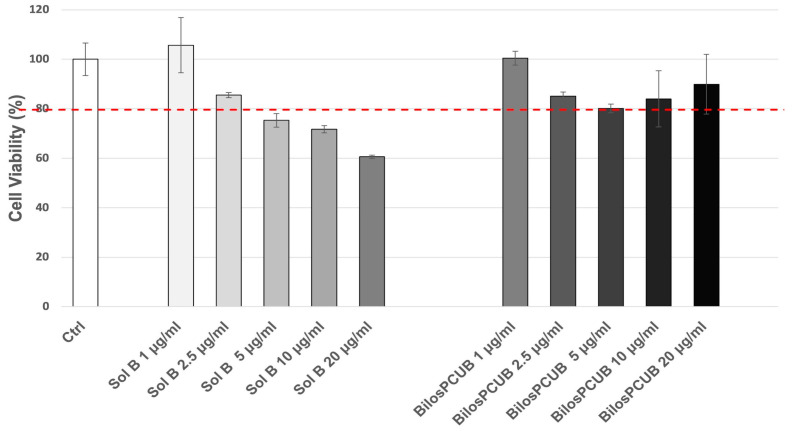
MTT assay on CaCo2 cells treated for 24 h with budesonide formulations (Sol B, Bilos_PCUB) at different doses ranging from 1 to 20 μg/mL. Cell viability is presented as percentage with respect to untreated control cells. Data are given as mean ± SD, representative of three independent experiments with at least three technical replicates each time. A value of 80% (red dotted line) was considered the reference accepted cell viability.

**Figure 8 antioxidants-12-02025-f008:**
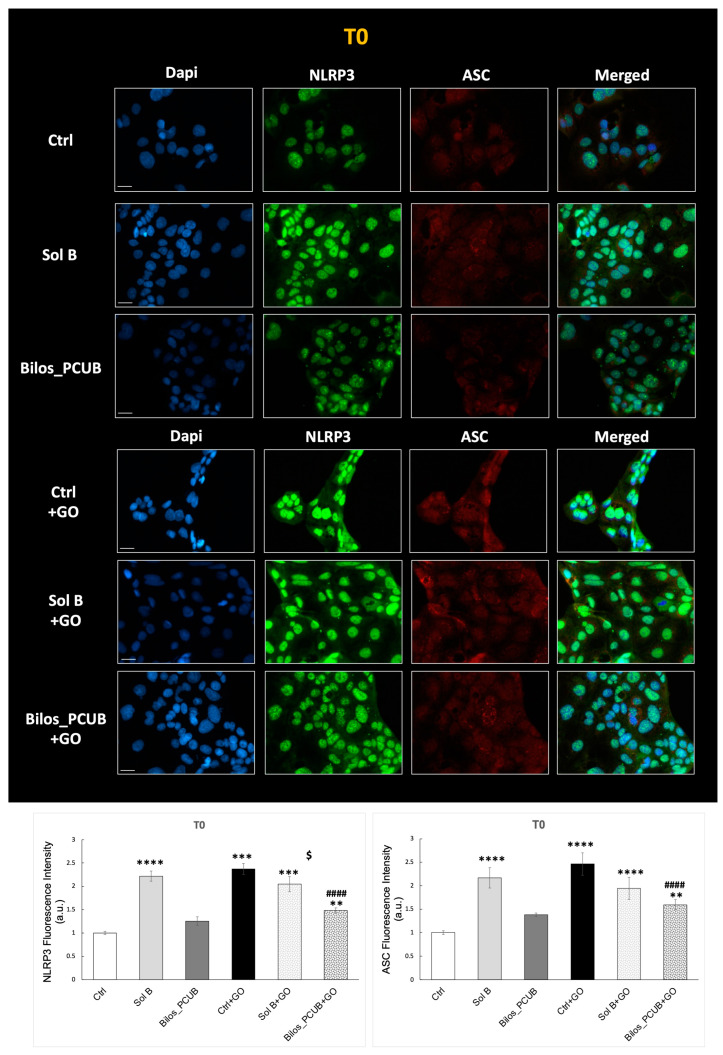
Immunofluorescence staining for NLRP3 (green) and ASC (red) on CaCo2 cells treated with formulations containing budesonide and exposed to 0.5 U/mL of GO for 1 h. Blue staining (DAPI) represents nuclei. Images were taken at 40 × magnification (scale bar = 40 μm), and the fluorescent signal was quantified using ImageJ software. Data are the results of the averages of at least three different experiments, obtained via 2-way ANOVA followed by Tukey’s post hoc comparison test. ^$^
*p* < 0.05; ** *p* < 0.01; *** *p* < 0.005; ****^,####^ *p* < 0.0001 (* with respect to Ctrl and # with respect to Ctrl + GO).

**Figure 9 antioxidants-12-02025-f009:**
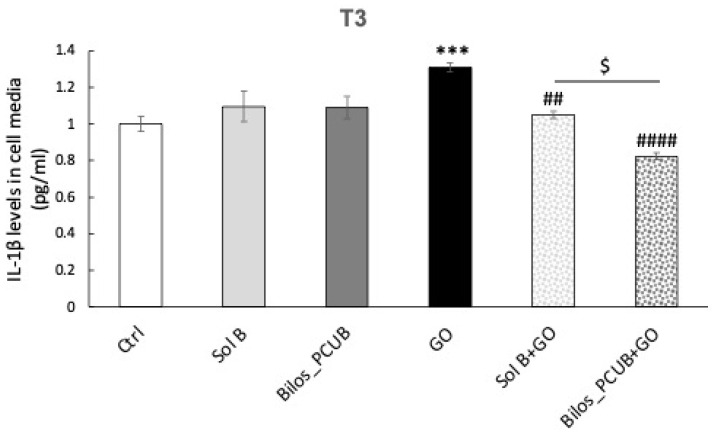
Released levels of IL- 1β in media of CaCo2 cells pre-treated with formulations containing budesonide, exposed to 0.5 U/mL of GO for 1 h and collected after 3 h (T3). Data are the results of the averages of at least three different experiments, obtained via 2-way ANOVA followed by Tukey’s post hoc comparison test. ^$^ *p* < 0.05; ^##^ *p* < 0.01; *** *p* < 0.005; ^####^ *p* < 0.0001 (* with respect to Ctrl and # with respect to Ctrl + GO).

**Table 1 antioxidants-12-02025-t001:** HPLC analysis conditions for samples containing Budesonide.

Column	Mobile Phase	Flow (mL/min)	Wavelength (nm)	Pressure (bar)	Retention Time (min)
Kinetex 5 μm C18 100 Å, RP, 150 × 4.6 mm	Acetonitrile and Phosphate buffer (pH 3.2) 60:40 *v/v*	0.8 (isocratic mode)	245	80	3.4–3.5

**Table 2 antioxidants-12-02025-t002:** Composition of receiving phases used in diffusion experiments.

Receiving Phase	Component	Concentration
FaSSGF (pH 1.6)	Sodium cholate Phosphatidylcholine NaCl	80 μM 20 μM 0.32 mM
FaSSIF (pH 6.5)	Sodium taurocholate Phosphatidylcholine porcine lipase	3.0 mM 0.2 mM 100 U/mL
PBS (pH 7.2–7.4)	NaH_2_PO_4_ × H_2_O NaHPO_4_ × 2 H_2_O NaCl	16.08 mM 80 mM 1.5 M

**Table 3 antioxidants-12-02025-t003:** Encapsulation efficiency of budesonide within the produced biloparticles.

Biloparticles	Encapsulation Efficiency * (%) ± s.d.
BilopS_CB	69.25 ± 6.48
BilopS_UB	59.10 ± 1.96
BilopS_TB	81.96 ± 2.76
BilopN_CB	61.87 ± 9.32
BilopN_UB	57.74 ± 1.66
BilopN_TB	70.20 ± 5.44

* Each value is the average of 5 batches ± standard deviation.

**Table 4 antioxidants-12-02025-t004:** Kinetic parameter R^2^ of budesonide released from bilosomes and biloparticles through synthetic and natural membranes.

Formulation	Nylon Membrane			Rat Intestinal Membrane		
	Zero Order	First Order	Second Order	Zero Order	First Order	Second Order
	F = K_0_ t	ln(1 − F) = −K_1_ t	F = K_2_ t^1/2^	F = K_0_ t	ln(1 − F) = −K_1_ t	F = K_2_ t^1/2^
Susp B	0.8753	0.8650	0.9878	0.9703	0.9642	0.9624
Bilos_PCUB	0.9038	0.9160	0.9614	0.9774	0.9584	0.9768
Bilos_PCCB	0.9365	0.9524	0.9850	0.9324	0.9128	0.9256
Bilos_PCTB	0.9634	0.9721	0.9763	0.9815	0.8605	0.9170
BilopS_UB	0.9798	0.9922	0.9601	0.8708	0.9343	0.9577
BilopS_CB	0.9717	0.9940	0.9808	0.8327	0.8776	0.9350
BilopS_TB	0.9922	0.9986	0.9589	0.9289	0.9721	0.9878
BilopN_UB	0.9807	0.9935	0.9531	0.8648	0.9305	0.9524
BilopN_CB	0.9461	0.9793	0.9720	0.9460	0.9643	0.9672
BilopN_TB	0.9632	0.9986	0.9572	0.8682	0.9315	0.9574

## Data Availability

Data are contained within the article and Appendix A.

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
