# Peer review of "Bilosomes and Biloparticles for the Delivery of Lipophilic Drugs: A Preliminary Study"

_antioxidants, 2023, doi:10.3390/antiox12122025_

Round 1

Reviewer 1 Report

Comments and Suggestions for Authors

The main objective of the study is to encapsulate a hydrophobic drug into lipid vesicles to improve its water solubility. The authors suggest taking advantage of the properties of bile acids-based vesicles and nanoparticles (i.e. bilosomes and biloparticles) to improve the water solubility of Budesonide.

Specific comments

- The advantage of using nanoparticles containing different bile acids in their composition, namely ursodeoxycholic acid (U), sodium cholate (C) and sodium taurocholate rather than PC-Cholesterol liposomes is questionable. How bile acids contribute to specific drug delivery properties is an open question that is never addressed.

- Figure 5 illustrates the budesonide release kinetics in FaSSGF and FaSSIF but does not provide information regarding the real amounts of released drug for each system.  It is a major concern; indeed relative % drug release does not allow to draw any conclusion regarding the advantage of using bilosomes to improve budesonide activity. Figure 5 should illustrate the changes in drug concentrations versus time.

- FASSGF simulates the composition and pH 1.6 of gastric juice. Is the significant improvement of budesonide passage into the aqueous solution a consequence of the biloparticles or bilosomes bilayer destabilization or a change of budenoside ionisation state. The same concern is about FASSIF which mimics the intestinal medium.

- After treatment with Bilos_PCUB but also Sol B the levels of IL-1β are fully restored at the basal state. What could be the advantage of a biloparticles or bilosomes treatment. Moreover, the study does not provide any experimental argument proving a direct effect on NLRP3 activity.

- The statement claiming that budesonide inserted into the bilayer makes the size of liposomes more homogeneous and monodisperse is questionable and not supported by any experimental argument.

Author Response

Dear Editor,

please find below the list of responses to the comments of Reviewer 1, made point-by-point.

Point 1- The advantage of using nanoparticles containing different bile acids in their composition, namely ursodeoxycholic acid (U), sodium cholate (C) and sodium taurocholate rather than PC-Cholesterol liposomes is questionable. How bile acids contribute to specific drug delivery properties is an open question that is never addressed.

Answer.

We thank the reviewer for this point. Maybe the specific drug delivery is a questionable open question, but in literature many studies reported that bile salt-phospholipid vesicles are able to transport the drug to the intestinal wall leading to increased bioavailability (Holm et al., Int J Pharm 453, 2013, 44-55; Zarenezhad et al., J Funct Biomater 2023, 14, 453). Furthermore, bile salts are physiologically involved in the digestion of lipids therefore they can affect the absorption of poorly soluble drugs in different ways. Especially if administered orally.

Bilosomes are described as nonionic, amphiphilic, flexible vesicles with bile salts integrated within, giving a self-assembling structure capable of overcoming some of liposome drawbacks by means of its high permeation potential and promising GIT enzymatic stability (Ahmad et al. Curr Pharm Des 2017, 23, 1575-1588; Mohsen et al. Drug Dev Ind Pharm 2017, 43,  2043-2054). The bilosomes’ stability and plasticity in the GIT facilitate the efficient carriage of drugs and bioactive compounds (Zarenezhad et al., J Funct Biomater 2023, 14, 453). In this contest the use of bilosomes (i.e. vesicles stabilized by bile salts) enable the delivery via the oral route protecting drug contents from the harsh environment of the gut and allowing a new opportunity to oral administration alternative to other vesicular delivery systems. 

Furthermore, the influence of different bile acids (i.e. U, C, T) on the performance of the produced bilosystem needs to be studied in depth to try to understand their contribution to the specific drug release properties. This study, in fact, being preliminary, focused mainly on the macroscopic evidence of the different behavior of the produced bilosystems. Subsequent studies will be carried out trying to delve deeper into the reasons for the different behaviors expressed by the different bile acids used.

Point 2- Figure 5 illustrates the budesonide release kinetics in FaSSGF and FaSSIF but does not provide information regarding the real amounts of released drug for each system.  It is a major concern; indeed relative % drug release does not allow to draw any conclusion regarding the advantage of using bilosomes to improve budesonide activity. Figure 5 should illustrate the changes in drug concentrations versus time.

Answer. 

As rightly indicated by the Reviewer, Figure 5 has been modified describing the variations of drug concentrations versus time. Concerning the advantage of bilosomes, to our knowledge the ability of a formulation to maintain the integrity of a drug increases the chances to achieve therapeutic tissue doses.  Based on the reviewer comments we believe that it is more appropriate to say that we obtain an improvement in the concentration and bioavailability of the drug capable of carrying out its therapeutic activity instead of an “improvement in activity”. Manuscript has been corrected accordingly.

Point 3- FASSGF simulates the composition and pH 1.6 of gastric juiceIs the significant improvement of budesonide passage into the aqueous solution a consequence of the biloparticles or bilosomes bilayer destabilization or a change of budenoside ionisation state. The same concern is about FASSIF which mimics the intestinal medium.

​​Answer. 

The reviewer's objection on this point is correct, however the answer is supported by the release results of Figure 5. Indeed when budesonide is in ethanol solution (not an administrable formulation) its release is probably very fast due to its ionization state in both types of simulated fluids. However, when budesonide is suspended in both simulated fluids (possible administrable formulation), its release remains very low even after 8 hours....why if its ionization occurs anyway? The answer perhaps lies in the fact that you need a tool that makes the drug a little more soluble in the system in order to be released and available for absorption. The presence of a bilosystems (either bilosome or biloparticle) is able to increase the solubility and to control the release of the incorporated drug. The overall results revealed significant bilosystem dominance to enhance intestinal passage of budesonide (Fig. 6). This can be explained by the ability of these nanocarriers to improve the intestinal permeability of the encapsulated drugs thanks to the membrane fluidizing effect that the bilosystems demonstrate due to the presence of the bile salts that compose them (Aditya et al, Food application. Biotechnol Adv 2017, 35, 450–457; Ahmed et al., Int J Nanomed 2022, 15, 9783; Binsuwaidan et al., Pharmaceuticals 2022, 15, 1043)

Point 4 - After treatment with Bilos_PCUB but also Sol B the levels of IL-1β are fully restored at the basal state. What could be the advantage of a biloparticles or bilosomes treatment. Moreover, the study does not provide any experimental argument proving a direct effect on NLRP3 activity.

Answer. 

Our results show that although Sol B can reduce IL-1β levels upon GO insult, indicating the effectiveness of the drug in protecting against the oxinflammatory damage, Bilos-PCUB displays a greater effect. These data suggest that Bilos-PCU can deliver budesonide better than Sol B (confirming what was stated in the answer to point 2), suggesting the promising use of this delivery system in the long term. In fact, in line with the formulation study, it is appropriate to think that Bilos_PCU allows a controlled release of budesonide compared to Sol B, thus allowing the release of the drug even after a prolonged period of time.

Considering that this preliminary study shows the promising role of Bilos-PCU as a delivery system, further studies are needed to better investigate the protective effect of Bilos_PCUB over a longer time (e.g, after 24 or 48 hours). 

Regarding the activity of NLRP3, the analysis of colocalization of NLRP3 with ASC, has been performed and added to the supplemental materials as new Figure S3. Indeed, when activated inflammasomes components tend to colocalize forming the active platform able to promote the release of inflammatory interleukins (Broz and Dixit, Nat Rev Immunol 2016, 16, 407-420).The data obtained show that the co-localization of NLRP3 and ASC occurs following GO insult, thus confirming that the inflammasome is active and that only Bilos-PCUB can significantly prevent the platform assembly, inhibiting the activation of the NLRP3 inflammasome.

Point 5 - The statement claiming that budesonide inserted into the bilayer makes the size of liposomes more homogeneous and monodisperse is questionable and not supported by any experimental argument.

Answer. 

We believe that the experimental results obtained in our study (Figure 2) clearly indicated that the presence of budesonide leads to more homogeneous and monodisperse vesicles’ dispersion. Even if this behavior was more evident for sodium cholate-containing vesicles as compared to those containing taurocholate or ursodeoxycholic acids. We tentatively hypothesized that the assembling of the drug within the bilayer could be responsible for this vesicle size reduction due to the interactions among the lipophilic moieties of phospholipids and bile acids. Some studies by Maritim et al, Mohammadi et al and Farzaneh et al, demonstrated that increasing amounts of cholesterol (a steroid molecule similar to budesonide) in the liposome bilayer improved particle uniformity (PDI) (Maritim et al.,  Int J Pharm 2021 5, 592, 120051; Mohammadi, et al. Curr Drug Deliv 2016, 13, 1065-1070; Farzaneh et al. Int J Pharm 2018, 551,  300-308). In addition Xian et al, presenting a study in which a synthetic budesonide derivative is assembled into liposomal nanovesicles for oral delivery to inflamed colon for IBD treatment, described that budesonide-loaded liposomes showed favorable colloidal stability in different media (i.e. PBS, FASSGF, FASSIF), with negligible variations in size and polydispersity within 1 week (Xian et al. Drug Delivery 2023, 30:1).

These considerations together (and the related literature) have been included in the manuscript to support the statement made in the text.

Thanking the Reviewer for the raised comments, we hope that now the manuscript could be accepted for publication.

Best regards

Rita Cortesi

Reviewer 2 Report

Comments and Suggestions for Authors

Thank you for providing an opportunity to review the manuscript titled “Bilosomes and biloparticles for the delivery of lipophilic drugs: a preliminary study”. After critical review of the manuscript, it can be accepted for publication after addressing the comments below.

Introduction:

Please add details on previous investigations where bilosomes or biloparticles were used to improve solubility of lipophilic drugs

Please mention the choice of budesonide and how these novel systems impact the bioavailability of budesonide in the body

Please provide rationale for choosing the particular lipids for this novel formulation

Conclusions

Please expand and add future studies planned

Author Response

Dear Editor,

please find below the list of responses to the comments of Reviewer 2, made point-by-point.

Point 1 - Please add details on previous investigations where bilosomes or biloparticles were used to improve solubility of lipophilic drugs

Answer. 

We thank the reviewer for this point, giving us the opportunity to improve the reference section of the manuscript. Indeed, iIn literature many studies are reported describing the use of bilosome (nonionic, amphiphilic, flexible vesicles  containing bile salts) to improve the solubility, permeability as well as the oral bioavailability of many drugs (Islam M et al., Pakistan J Pharm Sci 2020, 33, 2301-2306; Saifi et al., J Drug Del Sci Technol 2020, 57, 101634; Elnaggar et al., Int J Pharm 2019, 564, 410-425; Arzani et al., Int J Nanomed 2015; 10, 4797-813; Elkomyet al., Drug Deliv 2022, 29, 2694–2704; Liu et al., Colloids Surf. A Physicochem. Eng. Asp. 2022, 654, 130055; Aburahma, M.H. Bile salts-containing vesicles: Promising pharmaceutical carriers for oral delivery of poorly water-soluble drugs and peptide/protein-based therapeutics or vaccines. Drug Deliv. 2016, 23, 1847–1867). Additionally, bile salts increase the bioavailability of the drugs via adsorption or permeation into epithelial barriers (Gupta,et al. in Systems of Nanovesicular Drug Delivery; Elsevier: Amsterdam, The Netherlands, 2022; pp. 293–309). At present, bilosomes are mainly used to deliver vaccines and other biological macromolecules (Ahmad, et al., Coll Surf B: Biointerfaces, 2021, 197, 111389; D'Elia et al. J Control Rel, 2019, 298, 202-212; Mohsen et al. J Drug Del Sci Technol, 2020, 59, 101910; Saifi et al., J Drug Deli Sci. Technol. 2020, 57, 101634; Wilkhu, J.S.; McNeil, S.E.; Anderson, D.E.; Perrie, Y. Characterization and optimization of bilosomes for oral vaccine delivery. J. Drug Target. 2013, 21, 291–299; Zarenezhad, E.; Marzi, M.; Abdulabbas, H.T.; Jasim, S.A.; Kouhpayeh, S.A.; Barbaresi, S.; Ahmadi, S.; Ghasemian, A. Bilosomes as Nanocarriers for the Drug and Vaccine Delivery against Gastrointestinal Infections: Opportunities and Challenges. J. Funct. Biomater. 2023, 14, 453; Paes Dutra et al., Microparticles and nanoparticles-based approaches to improve oral treatment of Helicobacter pylori infection Crit Rev Microbiol. 2023 Oct 28:1-22; Célia Faustino, Cláudia Serafim, Patrícia Rijo & Catarina Pinto Reis (2016) Bile acids and bile acid derivatives: use in drug delivery systems and as therapeutic agents, Expert Opinion on Drug Delivery, 13:8, 1133-1148, DOI: 10.1080/17425247.2016.1178233; Wenjie Yao, Zhishi Xu, Jiang Sun, Jingwen Luo, Yinghui Wei, Jiafeng Zou,Deoxycholic acid-functionalised nanoparticles for oral delivery of rhein, European Journal of Pharmaceutical Sciences, Volume 159, 2021,105713; Mooranian A, Raj Wagle S, Kovacevic B, Takechi R, Mamo J, Lam V, Watts GF, Mikov M, Golocorbin-Kon S, Stojanovic G, Al-Sallami H, Al-Salami H. Bile acid bio-nanoencapsulation improved drug targeted-delivery and pharmacological effects via cellular flux: 6-months diabetes preclinical study. Sci Rep. 2020 Jan 9;10(1):106).

Point 2 - Please mention the choice of budesonide and how these novel systems impact the bioavailability of budesonide in the body

Answer. 

The choice of Budesonide was made on the basis  that it is considered the anti-inflammatory drug of "first choice" in the treatment of patients with mild to moderate inflammatory bowel disease (IBD) (Kozuch and Hanauer, World J Gastroenterol 2008 14, 354-77; Molodecky et al. Gastroenterology 2012, 142: 46-54; Hanauer, Gut 2002, 51, 182–3). Like all poorly water-soluble drugs, the corticosteroid Budesonide when administered orally is difficult to absorb in the gastrointestinal tract, resulting in poor bioavailability (Martinez and AmidonJ Clin Pharmacol 2002, 42, 620-43; Stegemann et al. Eur J Pharm Sci 2007, 31, 249-61), poor release at sites of intestinal inflammation and therefore low effectiveness. Furthermore, it is necessary to take into account the harsh conditions of the digestive tract which prevent an effective accumulation of drugs in intact form in inflamed intestinal sites (Lautenschläger et al., Adv Drug Deliv Rev 2014, 71, 58-76; Liu et al. Drug Deliv 2017, 24, 569-81) but also a high rate of metabolization and hepatic clearance (Derendorf and Meltzer Allergy 2008, 63, 1292-300). Therefore, strategies that improve patient compliance and also allow for high efficiency of administration through the gastrointestinal tract (Vrettos et al., Pharmaceutics 2021, 13,1591; Xian et al. Drug Delivery 2023, 30,1), such as bilosomes and biloparticles become interesting in order to expand the clinical use of this drug (Xian et al. Drug Delivery 2023, 30, 1). These considerations have now been included in the manuscript.

Point 3 - Please provide rationale for choosing the particular lipids for this novel formulation

Answer. 

The aim of this study is to investigate the ability of formulations in improving the solubility and the intestinal absorption of budesonide, thanks to the presence of bile acids. To address this point, lipids for the production of bilosystems were selected on the basis of the most common components used in preparation of traditional lipid-based nanosystems such as liposomes, SLN and NLC, as reported in literature (Mu and Holm, Expert Opin Drug Deliv 2018, 15(8), 771-785; Muller et al., Curr Drug Discov Technol 2011, 8(3), 207-27; da Silva Santos et al, Food Research International, 2019, 122, 610-626; Talegaonkar et al., AAPS PharmSciTech 2019, 20, 121; Maritim et al., Int J Pharm 2021 5, 592:120051). 

Point 4 - Conclusions Please expand and add future studies planned

Answer. 

We thank the reviewer for the suggestion. Conclusions in the manuscript have been implemented with future studies planned as suggested.

Thanking the Reviewer for the raised comments, we hope that now the manuscript could be accepted for publication.

Best regards

Rita Cortesi

Reviewer 3 Report

Comments and Suggestions for Authors

This manuscript presents a study focused on the development and characterization of bilosomes and biloparticles intended for the targeted delivery of lipophilic drugs. Since drug delivery can be challenging for several reasons, including barriers, size, charge, or chemical stability, it is meaningful to have a research article exploring the transport based on bilosomes and biloparticles. Suggestions are below.

(*) Table S1 presents the composition of biloparticles using the percentage by weight of every sample; why you have just 99.9% for BilopS and for BilopN. The other two samples contains 0.1% Budesonide, but here it is absent.

(*) 2.4 Dimensional Analysis

Samples dilution was 1:10; it is not clear if the ratio is volume / volume

It is also important to mention the solubility of your carriers in water; did you managed to dissolve them completely using this dissolution ratio?

 (*) 2.8 Mathematical and Statistical analysis

Did you tested the normality of data sets: Shapiro–Wilk (n<50) or Kolmogorov–Smirnov test (n≥50); you must also pay great attention to the choice of tests (parametric vs. non-parametric) based on the size of n

(*) 3.1.1. Bilosomes

You talk about the formation of aggregates, but you don't use Zetasizer Nano S90 to measure the samples Zeta potentials; why? The formation of clusters can be appreciated by surface charges and SEM

(*) Figure S2

Please use black letters and numbers also to PDI scale; it is much easier to read

(*) 3.2.1. Bilosomes

A formula to calculate EE% is better than a description in text

(*) Figure 4

Please use black letters and numbers also to PDI scale; it is much easier to read

(*) Figure 6

I don't know why, but I couldn't find this figure, only its title in the manuscript at page 12 of 18

(*) Figure 8

Please put the two graphs under the image with the immunofluorescence staining because they are too small and their axis cannot be read now

(*) References

Attention to the age of bibliography; only 23% of references are from the last 5 years

Author Response

Dear Editor,

please find below the list of responses to the comments of Reviewer 3, made point-by-point.

Point 1 -  Table S1 presents the composition of biloparticles using the percentage by weight of every sample; why you have just 99.9% for BilopS and for BilopN. The other two samples contains 0.1% Budesonide, but here it is absent.

Answer. 

We apologize for the  mistake. The amount of lipid used in empty BilopS and BilopN have been corrected in Table S1. As reported in section 2.3, the lipid phase represents the 5% of the total weight of the formulation, and the drug was added in the lipid phase. 

Point 2 -  2.4 Dimensional Analysis

Samples dilution was 1:10; it is not clear if the ratio is volume / volume. It is also important to mention the solubility of your carriers in water; did you managed to dissolve them completely using this dissolution ratio?

Answer. 

We thank the reviewer for these points. For the dimensional analysis, the dispersion samples were diluted 1:10 by volume and the manuscript was updated accordingly. Concerning  the second point, the produced nanosystems are lipid-based as reported in the production method, thus the dilution in water, selected to operate in an appropriate concentration range of vesicles or particles for the dimensional detection, does not dissolve the dispersed phase. 

Point 3 - 2.8 Mathematical and Statistical analysis

Did you tested the normality of data sets: Shapiro–Wilk (n<50) or Kolmogorov–Smirnov test (n≥50); you must also pay great attention to the choice of tests (parametric vs. non-parametric) based on the size of n

Answer. 

We thank the reviewer for the suggestion. However, we firstly tested the normality of data sets by assessing a Shapiro-Wilk test. Considering that the samples were normally distributed, the further statistical analysis was conducted following a parametric test.

Now, this specification has been included within the manuscript.

Point 4 - 3.1.1. Bilosomes

You talk about the formation of aggregates, but you don't use Zetasizer Nano S90 to measure the samples Zeta potentials; why? The formation of clusters can be appreciated by surface charges and SEM

Answer. 

We thank the Reviewer for this suggestion. Unfortunately our Zetasizer Nano is an S90 and not a ZS90, so we are unable to measure the Zeta potential and follow the electrical charge variation of the dispersed phase. However, since this was a preformulation study and aggregate formation was evident after one month of production only for dispersions prepared using a hydrating solution at room temperature, we were able to discard this hydration method and opt for a film hydration with a hot solution. Finally, regarding SEM visualization, it is not at all a good method for seeing vesicles in aqueous dispersion and thus it makes no sense to consider it. Furthermore, since the formulations are dispersed systems, Cryo-TEM analysis has been conducted and images have been shown in Figures 1 and 3.

Point 5 - Figure S2

Please use black letters and numbers also to PDI scale; it is much easier to read 

Answer. 

Figure S2 has been updated as requested.

Point 6 - 3.2.1. Bilosomes

A formula to calculate EE% is better than a description in text

Answer. 

In order to follow the suggestion of the Reviewer, the formula  to calculate EE% has been included in the manuscript.

Point 7 - Figure 4

Please use black letters and numbers also to PDI scale; it is much easier to read

Answer. 

Figure 4 has been updated as requested.

Point 8 - Figure 6

I don't know why, but I couldn't find this figure, only its title in the manuscript at page 12 of 18

Answer. 

Yes, the Reviewer is right, figure 6 in this version was inexplicably missed. Now the 

Point 9 - Figure 8

Please put the two graphs under the image with the immunofluorescence staining because they are too small and their axis cannot be read now

Answer. 

We thank the reviewer for the suggestion. In the revised version of the manuscript the figure has been updated as requested .

Point 10 - References

Attention to the age of bibliography; only 23% of references are from the last 5 years

Answer. 

We thank the reviewer for flagging this issue. We updated the reference in number and publication year.

Thanking the Reviewer for the raised comments, we hope that now the manuscript could be accepted for publication.

Best regards

Rita Cortesi

Round 2

Reviewer 1 Report

Comments and Suggestions for Authors

.

Reviewer 3 Report

Comments and Suggestions for Authors

I checked with interest all the changes made by you and I consider that the manuscript is now in a form in which it can be published